# Learning temporal attention in dynamic graphs with bilinear interactions

**Boris Knyazev**[1,2☯]*, **Carolyn Augusta**[2,3,4☯], **Graham W. Taylor**[1,2,5]

**1** School of Engineering, University of Guelph, Guelph, Ontario, Canada, **2** Vector Institute for Artificial Intelligence, Toronto, Ontario, Canada, **3** Department of Mathematics & Statistics, University of Guelph, Guelph, Ontario, Canada, **4** Edwards School of Business, University of Saskatchewan, Saskatoon, Saskatchewan, Canada, **5** Canada CIFAR AI Chair, Guelph, Canada

☯ These authors contributed equally to this work.
* bknyazev@uoguelph.ca

**Data Availability Statement:** We use two publicly available data sets: the GitHub Archive (available at: https://www.gharchive.org) and the Social Evolution data set (available at:

## Abstract

Reasoning about graphs evolving over time is a challenging concept in many domains, such as bioinformatics, physics, and social networks. We consider a common case in which edges can be short term interactions (e.g., messaging) or long term structural connections (e.g., friendship). In practice, long term edges are often specified by humans. Human-specified edges can be both expensive to produce and suboptimal for the downstream task. To alleviate these issues, we propose a model based on temporal point processes and variational autoencoders that learns to infer temporal attention between nodes by observing node communication. As temporal attention drives between-node feature propagation, using the dynamics of node interactions to learn this key component provides more flexibility while simultaneously avoiding issues associated with human-specified edges. We also propose a bilinear transformation layer for pairs of node features instead of concatenation, typically used in prior work, and demonstrate its superior performance in all cases. In experiments on two datasets in the dynamic link prediction task, our model often outperforms the baseline model that requires a human-specified graph. Moreover, our learned attention is semantically interpretable and infers connections similar to actual graphs.

## Introduction

Graph structured data arise from fields as diverse as social network analysis, epidemiology, finance, and physics, among others [1–4]. A graph $\mathcal{G} = (\mathcal{V}, \mathcal{E})$ is comprised of a set of $N$ nodes, $\mathcal{V}$, and the edges, $\mathcal{E}$, between them. For example, a social network graph may consist of a set of people (nodes), and the edges may indicate whether two people are friends. Recently, graph neural networks (GNNs) [5–10] have emerged as a key modeling technique for learning representations of such data. These models use recursive neighborhood aggregation to learn latent features, $Z^{(t)} \in \mathbb{R}^{N \times c}$, of the nodes at state $t$, given features, $Z^{(t-1)} \in \mathbb{R}^{N \times d}$, at the previous state, $t - 1$:

$$Z^{(t)} = f(Z^{(t-1)}, A, W^{(t)}), \tag{1}$$

http://realitycommons.media.mit.edu/socialevolution4.html).

**Funding:** BK is funded by the Mila internship, the Vector Institute, the University of Guelph and DARPA (FA8750-17-C-0115). CA is funded by the University of Saskatchewan. GWT is funded by CIFAR, Canada Research Chairs and the University of Guelph. For a portion of time during this study, GWT received salary from a commercial company: Google. The funder provided support in the form of salary for GWT, but did not have any additional role in the study design, data collection and analysis, decision to publish, or preparation of the manuscript. Specifically, during this time GWT worked on this study in his capacity as a professor at the University of Guelph. This appointment was ongoing during his sabbatical at Google. The specific roles of these authors are articulated in the 'author contributions' section. The views, opinions and/or findings expressed are those of the authors and should not be interpreted as representing the official views or policies of the Department of Defense or the U.S. Government. The authors also acknowledge equipment support from Canada Foundation for Innovation. Resources used in preparing this research were provided, in part, by the Province of Ontario, the Government of Canada through CIFAR, and companies sponsoring the Vector Institute: http://www.vectorinstitute.ai/#partners.

**Competing interests:** For a portion of time during this study, GWT received salary from a commercial company: Google. This does not alter our adherence to PLOS ONE policies on sharing data and materials. Specifically, during this time GWT worked on this study in his capacity as a professor at the University of Guelph. This appointment was ongoing during his sabbatical at Google.

where $A \in \mathbb{R}^{N \times N}$ is an adjacency matrix of graph $\mathcal{G}$. $W^{(t)}$ are trainable parameters; $d$, $c$ are input and output dimensionalities, and $f$ is some differentiable function, which is typically an aggregation operator followed by a nonlinearity. State $t$ can correspond to either a layer in a feedforward network or a time step in a recurrent net [6].

The focus of GNNs thus far has been on static graphs [1]— graphs with a fixed adjacency matrix $A$. However, a key component of network analysis is often to predict the state of a graph as $A$ evolves over time. For example, knowledge of the evolution of person-to-person interactions during an epidemic facilitates analysis of how a disease spreads [11], and can be expressed in terms of the links between people in a dynamic graph. Other applications include predicting friendship, locations of players or some interaction between them in team sports, such as soccer [12, 13].

In many dynamic graph applications, edges can be: 1) short term (and usually frequent) interactions, such as direct contacts, messaging, passes in sports, or 2) long term intrinsic connections, such as sibling connections, affiliations, and formations. In practice, these long term edges are often specified by humans, and can be suboptimal and expensive to obtain. The performance of methods such as DyRep [14] rely heavily on the quality of long term edge information.

To alleviate this limitation, we take a different approach and infer long term structure jointly with the target task. The inferred structure is modelled as temporal attention between edges (Fig 1). We use DyRep [14] as a backbone model and Neural Relational Inference (NRI) [15] to learn attention.

We also propose a bilinear transformation layer for pairs of node features instead of concatenation as typically employed in prior work, including DyRep and NRI. On two dynamic graph datasets, Social Evolution [16] and GitHub (https://www.gharchive.org/), we achieve strong performance on the task of dynamic link prediction, and show interpretability of learned temporal attention sliced at particular time steps.

# 1 Related work

Prior work [13, 15, 17–21] addressing the problem of learning representations of entire dynamic graphs has tended to develop methods that are very specific to the application, with only a few shared ideas. This is primarily due to the difficulty of learning from temporal data in general and temporal graph-structured data in particular, which remains an open problem [3] that we address in this work.

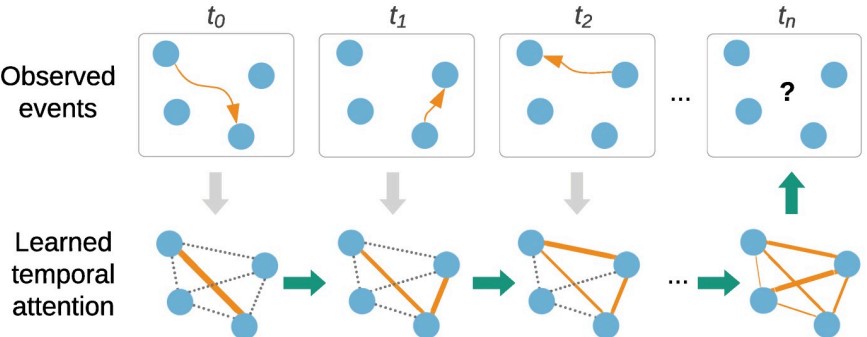

**Fig 1. A simple example illustrating the goal of our work, which is to learn temporal attention between nodes by observing the dynamics of events.** The learned attention is then used to make better future predictions. Dotted edges denote attention values yet to be updated.

**RNN-based methods**. Given an evolving graph $[\ldots, \mathcal{G}_{t-1}, \mathcal{G}_t, \mathcal{G}_{t+1}, \ldots]$, where $t \in [0, T-1]$ is a discrete index at a continuous time point $\tau$, most prior work uses some variant of a recurrent neural network (RNN) to update node embeddings over time [13, 15, 19, 21, 22]. The weights of RNNs are typically shared across nodes [15, 19, 21, 22]; however, in the case of smaller graphs, a separate RNN per-node can be learned to better tune the model [13]. RNNs are then combined with some function that aggregates features of nodes. While this can be done without explicitly using GNNs (e.g., using the sum or average of all nodes instead), GNNs impose an inductive relational bias specific to the domain, which often improves performance [13, 15, 19, 21].

**Temporal knowledge graphs**. The problem of inferring missing links in a dynamic graph is analogous to the problem of inferring connections in a temporal knowledge graph (TKG). In a TKG, nodes and labelled edges represent entities and the relationships between them, respectively [23]. Each edge between two nodes encodes a real-world fact that can be represented as a tuple, such as: (Alice, Liked, Chopin, 2019). In this example, Alice and Chopin are the nodes, the fact that Alice liked Chopin is the relationship, and 2019 is the timestamp associated with that fact. It may be that Alice no longer likes Chopin, as tastes change over time. To allow these timestamps to inform link prediction, previous work has provided a score for the entire tuple together [23], or by extending static embedding methods to incorporate the time step separately [24–27]. These methods tend to focus on learning embeddings of particular relationships, instead of learning the entire TKG.

**Dynamic link prediction**. There are several recent papers that have explored dynamic link prediction. In DynGem [28], graph autoencoders are updated in each time step. The skip-gram model in [29] incorporates temporal random walks. Temporal attention is revisited in DySAT [30], whereas [31] learn the next graph in a sequence based on a reconstruction error term. DynamicTriad [32] uses a triad closure to learn temporal node embeddings. LSTMs are featured in both GC-LSTM [33] and DyGGNN [34], and both involve graph neural networks, although only the latter uses whole-graph encoding.

DyRep [14] is a method that has been proposed for learning from dynamic graphs based on temporal point processes [35]. This method:

- supports two time scales of graph evolution (i.e. long-term and short-term edges);

- operates in continuous time;

- scales well to large graphs; and

- is data-driven due to employing a GNN similar to (1).

These key advantages make DyRep favorable, compared to other methods discussed above.

Another interesting approach to the modeling of dynamic graphs is based on the neural Hawkes processes [36]. However, it is limited to a few event types, which was addressed only recently in [37]. This approach is similar to DyRep, since it also relies on the temporal point processes and uses a neural network to learn dynamic representation. In [14], DyRep was shown to outperform a particular version of the Hawkes processes, which further supports our choice of using DyRep as the baseline. We leave a detailed comparison between a more recent version of the Hawkes processes [37] and DyRep for future work.

Finally, closely related to our work, there are a few applications where graph $\mathcal{G}_t$ is considered to be either unknown or suboptimal, so it is inferred simultaneously with learning the model [15, 19, 38]. Among them, [19, 38] focus on visual data, while NRI [15] proposes a more general framework and, therefore, is adopted in this work. For a review on other approaches for learning from dynamic graphs we refer to [39].

## 2 Background: DyRep

Here we describe relevant details of the DyRep model. A complete description can be found in [14]. DyRep is a representation framework for dynamic graphs evolving according to two elementary processes:

- $k = 0$: **Long-term association**, in which nodes and edges are added or removed from the graph affecting the evolving adjacency matrix $A^t \in \mathbb{R}^{N \times N}$.

- $k = 1$: **Communication**, in which nodes communicate over a short time period, whether or not there is an edge between them.

For example, in a social network, association may be represented as one person adding another as a friend. A communication event may be an SMS message between two friends (an association edge exists) or an SMS message between people who are not friends yet (an association edge does not exist). These two processes interact to fully describe information transfer between nodes. Formally, an event is a tuple $o^t = (u, v, \tau, k)$ of type $k \in \{0, 1\}$ between nodes $u$ and $v$ at continuous time point $\tau$ with time index $t$. For example, the tuple $o^1 = (1, 3, 9{:}10\text{AM}, 1)$ would indicate that node $u = 1$ communicates with node $v = 3$ at time point $t = 1$ corresponding to $\tau = 9{:}10$ in the morning.

### 2.1 Node update

Each event between any pair of nodes $u$, $v$ triggers an update of node embeddings $\mathbf{z}_u^t, \mathbf{z}_v^t \in \mathbb{R}^d$ followed by an update of temporal attention $S^t \in \mathbb{R}^{N \times N}$ in a recursive way (Fig 2). That is, updated node embeddings affect temporal attention, which in turn affects node embeddings at the next time step. In particular, the embedding of node $v$, and analogously node $u$, is updated based on the following three terms:

$$\mathbf{z}_v^t = \sigma\big[\underbrace{W^{\mathrm{S}}\mathbf{h}_u^{\mathrm{S},t-1}}_{\textbf{Attention-based}} + \underbrace{W^{\mathrm{R}}\mathbf{z}_v^{t_v-1}}_{\text{Self-propagation}} + \underbrace{W^{\mathrm{T}}(\tau - \tau^{t_v-1})}_{\text{Temporal shift}}\big], \tag{2}$$

where $W^{\mathrm{S}} \in \mathbb{R}^{d \times d}$, $W^{\mathrm{R}} \in \mathbb{R}^{d \times d}$ and $W^{\mathrm{T}} \in \mathbb{R}^d$ are learned parameters and $\sigma$ is a nonlinearity.

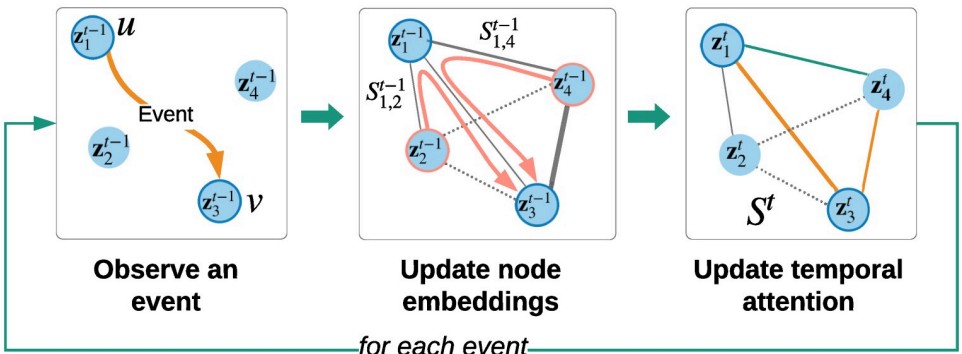

**Fig 2. A recursive update of node embeddings $z$ and temporal attention $S$.** An event between nodes $u = 1$ and $v = 3$ creates a temporary edge that allows the information to flow (in pink arrows) from neighbors of node $u$ to node $v$. Orange edges denote updated attention values. Normalization of attention values to sum to one can affect attention of neighbors of node $v$. Dotted edges denote absent edges (or edges with small values).

**Attention-based term.** We highlight the importance of the first term, which is affected by temporal attention $S^{t-1}$ between node $u$ and all its one-hop neighbors $\mathcal{N}_u^{t-1}$ [14]:

$$\mathbf{h}_u^{S,t-1} = f[\text{softmax}(S_u^{t-1})_i \cdot (W^h \mathbf{z}_i^{t-1}), \forall i \in \mathcal{N}_u^{t-1}], \tag{3}$$

where $f$ is an aggregator and $W^h \in \mathbb{R}^{d \times d}$ are learned parameters. Note that features of node $u$'s neighbors are used to update node $v$'s embedding, which can be interpreted as creating a temporal edge by which node features propagate between the two nodes. The amount of information propagated from node $u$'s neighbors is controlled by attention $S^{t-1}$, which we suggest to learn as described in Section 3.

The other two terms in (2) include a recurrent update of node $v$'s features and a function of the waiting time between the current event and the previous event involving node $v$.

## 2.2 Attention update

In DyRep, attention $S^t$ between nodes $u$, $v$ depends on three terms: 1) long term associations $A^{t-1}$; 2) attention $S^{t-1}$ at the previous time step; and 3) *conditional intensity* $\lambda_{k,u,v}^t$ (for simplicity denoted as $\lambda_k^t$ hereafter) of events between nodes $u$ and $v$, which in turn depends on their embeddings:

$$S_{uv}^t = f_S(A^{t-1}, S^{t-1}, \lambda_k^t), \tag{4}$$

where $f_S$ is Algorithm 1 in [14]. Let us briefly describe the last term.

**Conditional intensity $\lambda$.** Conditional intensity $\lambda_k^t$ represents the instantaneous rate at which an event of type $k$ (i.e., association or communication) occurs between nodes $u$ and $v$ in the infinitesimally small interval $(\tau, \tau + \delta\tau]$ [40]. DyRep formulates the conditional intensity as a softplus function of the concatenated learned node representations $\mathbf{z}_u^{t-1}, \mathbf{z}_v^{t-1} \in \mathbb{R}^d$:

$$\lambda_k^t = \psi_k \log\left(1 + \exp\left\{\frac{\omega_k^\top [\mathbf{z}_u^{t-1}, \mathbf{z}_v^{t-1}]}{\psi_k}\right\}\right), \tag{5}$$

where $\psi_k$ is the scalar trainable rate at which events of type $k$ occur, and $\omega_k \in \mathbb{R}^{2d}$ is a trainable vector that represents the compatibility between nodes $u$ and $v$ at time $t$; $[\cdot, \cdot]$ is the concatenation operator. Combining (5) with (4), we can notice that $S_{u,v}^t$ implicitly depends on the embeddings of nodes $u$ and $v$.

In DyRep, $\lambda$ is mainly used to compute the loss and optimize the model, so that its value between nodes involved in the events should be high, whereas between other nodes it should be low.

But $\lambda$ also affects attention values using a hard-coded algorithm, which makes several assumptions (see ALGORITHM 1 in [14]). In particular, in the case of a communication event ($k = 1$) between nodes $u$ and $v$, attention $S^t$ is only updated if an association exists between them ($A_{u,v}^{t-1} = 1$). Secondly, to compute attention between nodes $u$ and $v$, only their one-hop neighbors at time $t - 1$ are considered. Finally, no learned parameters directly contribute to attention values, which can limit information propagation, especially in case of imperfect associations $A^{t-1}$.

In this paper, we extend the DyRep model in two ways. First, we examine the benefits of learned attention instead of the DyRep's algorithm in (4) by using a variational autoencoder, more specifically its encoder part, from NRI [15]. This permits learning of a sparse representation of the interactions between nodes instead of using a hard-coded function $f_S$ and known adjacency matrix $A$ (Section 3.1). Second, both the original DyRep work [14] and [15] use concatenation to make predictions for a pair of nodes (see (5) above and (6) in [15]), which only

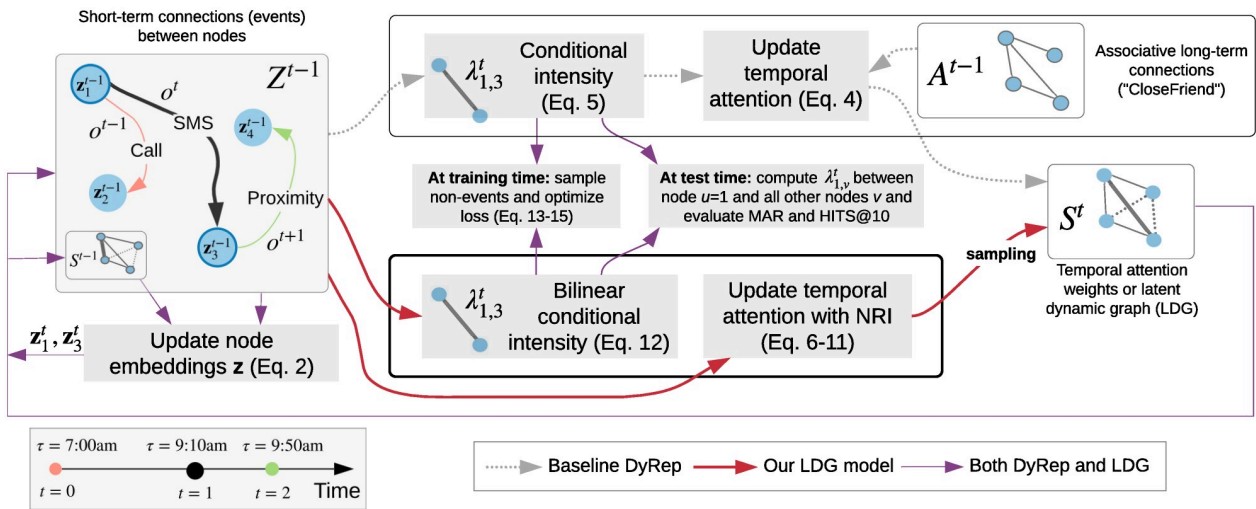

**Fig 3. Overview of our approach relative to DyRep [14], in the context of dynamic link prediction.** During training, events $o^t$ are observed, affecting node embeddings $Z$. In contrast to DyRep, which updates attention weights $S^t$ in a predefined hard-coded way based on associative connections $A^t$, such as CLOSEFRIEND, we assume that graph $A^t$ is unknown and our latent dynamic graph (LDG) model based on NRI [15] infers $S^t$ by observing how nodes communicate. We show that learned $S^t$ has a close relationship to certain associative connections. Best viewed in colour.

captures relatively simple relationships. We are interested in the effect of allowing a more expressive relationship to drive graph dynamics, and specifically to drive temporal attention (Sections 3.1 and 3.2). We demonstrate the utility of our model by applying it to the task of dynamic link prediction on two graph datasets (Section 4).

## 3 Latent dynamic graph model

In [15], Neural Relational Inference (NRI) was proposed, showing that in some settings, models that use a learned representation, in which the human-specified graph structure is discarded, can outperform models that use the explicit graph. A learned sparse graph representation can retain the most salient features, i.e., only those connections that are necessary for the downstream task, whereas the human-specified graph may have redundant connections.

While NRI learns the latent graph structure from observing node movement, we learn the graph by observing how nodes communicate. In this spirit, we repurpose the encoder of NRI, combining it with DyRep, which gives rise to our latent dynamic graph (LDG) model, described below in more detail (Fig 3). We also summarize our notation in Table 1.

### 3.1 Bilinear encoder

DyRep's encoder (4) requires a graph structure represented as an adjacency matrix $A$. We propose to replace this with a sequence of learnable functions $f_S^{\text{enc}}$, borrowed from [15], that only require node embeddings as input:

$$S^t = f_S^{\text{enc}}(Z^{t-1}). \tag{6}$$

Given an event between nodes $u$ and $v$, our encoder takes the embedding of each node $j \in \mathcal{V}$ at the previous time step $\mathbf{z}_j^{t-1}$ as an input, and returns an edge embedding $\mathbf{h}_{(u,v)}^2$ between nodes $u$

**Table 1. Mathematical symbols used in model description. Number of relation types $r = 1$ in DyRep and $r > 1$ in the proposed LDG.**

| Notation | Dimensionality | Definition |
|---|---|---|
| $\tau$ | 1 | Point in continuous time |
| $t$ | 1 | Time index |
| $t_v{-}1$ | 1 | Time index of the previous event involving node $v$ |
| $\tau^t_v{-}1$ | 1 | Time point of the previous event involving node $v$ |
| $i, j$ | 1 | Index of an arbitrary node in the graph |
| $v, u$ | 1 | Index of a node involved in the event |
| $A^t$ | $N \times N$ | Adjacency matrix at time $t$ |
| $\mathcal{N}_u(t)$ | $\lvert \mathcal{N}_u(t) \rvert$ | One-hop neighbourhood of node $u$ |
| $Z^t$ | $N \times d$ | Node embeddings at time $t$ |
| $\mathbf{z}^t_v$ | $d$ | Embedding of node $v$ at time $t$ |
| $\mathbf{h}^{S,t-1}_u$ | $d$ | Attention-based embedding of node $u$'s neighbors at time $t - 1$ |
| $\mathbf{h}^1_j$ | $d$ | Learned hidden representation of node $j$ after the first pass |
| $\mathbf{h}^1_{i,j}$ | $d$ | Learned hidden representation of an edge between nodes $i$ and $j$ after the first pass |
| $\mathbf{h}^2_j$ | $d$ | Learned hidden representation of node $j$ after the second pass |
| $\mathbf{h}^2_{(u,v)}$ | $r$ | Learned hidden representation of an edge between nodes $u$ and $v$ involved in the event after the second pass (Fig 4) |
| $S^t$ | $N \times N \times r$ | Attention at time $t$ with $r$ multirelational edges |
| $S^t_u$ | $N \times r$ | Attention between node $u$ and its one-hop neighbors at time $t$ |
| $S^t_{u,v}$ | $r$ | Attention between nodes $u$ and $v$ at time $t$ for all edge types $r$ |
| $\lambda^t_k$ | 1 | Conditional intensity of edges of type $k$ at time $t$ between nodes $u$ and $v$ |
| $\psi_k$ | 1 | Trainable rate at which edges of type $k$ occur |
| $\omega_k$ | $2d$ | Trainable compatibility of nodes $u$ and $v$ at time $t$ |
| $\Omega_k$ | $d \times d$ | Trainable bilinear interaction matrix between nodes $u$ and $v$ at time $t$ |

and $v$ using two passes of node and edge mappings, denoted by the superscripts $^1$ and $^2$ (Fig 4):

$$1^{st} \text{ pass} \begin{cases} \forall j : \mathbf{h}^1_j & = f^1_{\text{node}}(\mathbf{z}^{t-1}_j) & (7) \\ \forall i, j : \mathbf{h}^1_{(i,j)} & = f^1_{\text{edge}}(\mathbf{h}^{1\top}_i W^1 \mathbf{h}^1_j) & (8) \end{cases}$$

$$2^{nd} \text{ pass} \begin{cases} \forall j : \mathbf{h}^2_j & = f^2_{\text{node}}(\sum_{i \neq j} \mathbf{h}^1_{(i,j)}) & (9) \\ u, v : \mathbf{h}^2_{(u,v)} & = f^2_{\text{edge}}(\mathbf{h}^{2\top}_u W^2 \mathbf{h}^2_v) & (10) \end{cases}$$

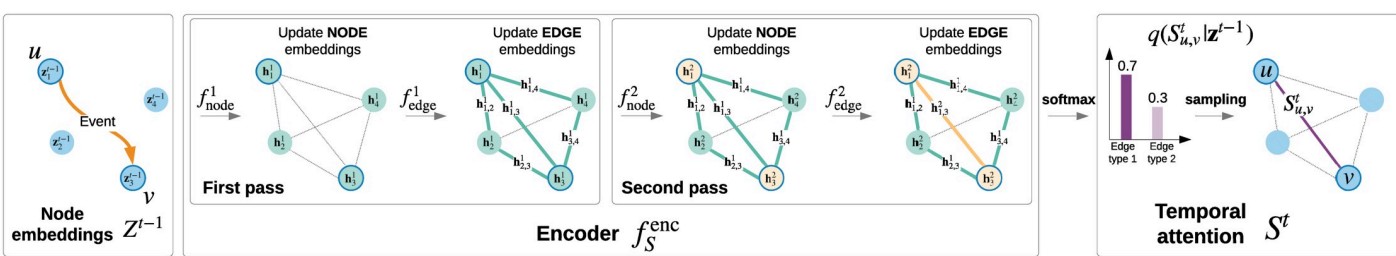

**Fig 4. Inferring an edge $S^t_{u,v}$ of our latentdynamic graph (LDG) using two passes, according to (7)–(10), assuming an event between nodes $u = 1$ and $v = 3$ has occurred.** Even though only nodes $u$ and $v$ have been involved in the event, to infer the edge $S^t_{u,v}$ between them, interactions with all nodes in a graph are considered.

where $f^1_{\text{node}}, f^1_{\text{edge}}, f^1_{\text{node}}, f^2_{\text{edge}}$ are two-layer, fully-connected neural networks, as in [15]; $W^1$ and $W^2$ are trainable parameters implementing bilinear layers. In detail:

- (7): $f^1_{\text{node}}$ transforms embeddings of all nodes;

- (8): $f^1_{\text{edge}}$ is a "node to edge" mapping that returns an edge embedding $\mathbf{h}^1_{(i,j)}$ for all pairs of nodes $(i, j)$;

- (9): $f^2_{\text{node}}$ is an "edge to node" mapping that updates the embedding of node $j$ based on its all incident edges;

- (10): $f^2_{\text{edge}}$ is similar to the "node to edge" mapping in the first pass, $f^1_{\text{edge}}$, but only the edge embedding between nodes $u$ and $v$ involved in the event is used.

The softmax function is applied to the edge embedding $\mathbf{h}^2_{(u,v)}$, which yields the edge type posterior as in NRI [15]:

$$q_\phi(S^t_{u,v}|Z^{t-1}) \equiv \text{softmax}(\mathbf{h}^2_{(u,v)}), \tag{11}$$

where $S^t_{u,v} \in \mathbb{R}^r$ are temporal one-hot attention weights sampled from the multirelational conditional multinomial distribution $q_\phi(S^t_{u,v}|Z^{t-1})$, hereafter denoted as $q_\phi(S|Z)$ for brevity; $r$ is the number of edge types (note that in DyRep $r = 1$); and $\phi$ are parameters of the neural networks in (7)–(10). $S^t_{u,v}$ is then used to update node embeddings at the next time step, according to (2) and (3).

Replacing (4) with (5) means that it is not necessary to maintain an explicit representation of the human-specified graph in the form of an adjacency matrix. The evolving graph structure is implicitly captured by $S^t$. While $S^t$ represents temporal attention weights between nodes, it can be thought of as a graph evolving over time, therefore we call our model a Latent Dynamic Graph (LDG). This graph, as we show in our experiments, can have a particular semantic interpretation.

The two passes in (7)–(10) are important to ensure that attention $S^t_{u,v}$ depends not only on the embeddings of nodes $u$ and $v$, $\mathbf{z}^{t-1}_u$ and $\mathbf{z}^{t-1}_v$, but also on how they interact with other nodes in the entire graph. With one pass, the values of $S^t_{u,v}$ would be predicted based only on local information, as only the previous node embeddings influence the new edge embeddings in the first pass (8). This is one of the core differences of our LDG model compared to DyRep, where $S^t_{u,v}$ only depends on $\mathbf{z}^{t-1}_u$ and $\mathbf{z}^{t-1}_v$ (see (4)).

Unlike DyRep, the proposed encoder generates multiple edges between nodes, i.e., $S^t_{u,v}$ are one-hot vectors of length $r$. We therefore modify the "Attention-based" term in (2), such that features $h^{S,t-1}_u$ are computed for each edge type and parameters $W^S$ act on concatenated features from all edge types, i.e., $W^S \in \mathbb{R}^{rd \times d}$.

### 3.2 Bilinear intensity function

Earlier in (5) we introduced conditional intensity $\lambda^t_k$, computed based on concatenating node embeddings. We propose to replace this concatenation with bilinear interaction:

$$\lambda^t_k = \psi_k \log\left(1 + \exp\left\{\frac{[\mathbf{z}^{t-1^\top}_u \mathbf{\Omega}_k \mathbf{z}^{t-1}_v]}{\psi_k}\right\}\right), \tag{12}$$

where $\mathbf{\Omega}_k \in \mathbb{R}^{d \times d}$ are trainable parameters that allows more complex interactions between evolving node embeddings.

**Why bilinear?** Bilinear layers have proven to be advantageous in settings like Visual Question Answering (e.g., [41]), where multi-modal embeddings interact. It takes a form of $\mathbf{x}W\mathbf{y}$, so that each weight $W_{ij}$ is associated with a pair of features $\mathbf{x}_i, \mathbf{y}_j$, where $i, j \in d$ and $d$ is the dimensionality of features $\mathbf{x}$ and $\mathbf{y}$. That is, the result will include products $\mathbf{x}_i W_{ij} \mathbf{y}_j$. In the concatenation case, the result will only include $\mathbf{x}_i W_i$, $\mathbf{y}_j W_j$, i.e. each weight is associated with a feature either only from $\mathbf{x}$ or from $\mathbf{y}$, not both. As a result, bilinear layers have some useful properties, such as separability [42]. In our case, they permit a richer interaction between embeddings of different nodes. So, we replace NRI's and DyRep's concatenation layers with bilinear ones in (8), (10) and (12).

### 3.3 Training the LDG model

Given a minibatch with a sequence of $\mathcal{P}$ events, we optimize the model by minimizing the following cost function:

$$\mathcal{L} = \mathcal{L}_{\text{events}} + \mathcal{L}_{\text{nonevents}} + \text{KL}[q_\phi(S|Z)||p_\theta(S)], \tag{13}$$

where $\mathcal{L}_{\text{events}} = -\sum_{p=1}^{\mathcal{P}} \log(\lambda_{k_p}^{t_p})$ is the total negative log of the intensity rate for all events between nodes $u_p$ and $v_p$ (i.e., all nodes that experience events in the minibatch); and $\mathcal{L}_{\text{nonevents}} = \sum_{m=1}^{\mathcal{M}} \lambda_{k_m}^{t_m}$ is the total intensity rate of all nonevents between nodes $u_m$ and $v_m$ in the minibatch. Since the sum in the second term is combinatorially intractable in many applications, we sample a subset of nonevents according to the Monte Carlo method as in [14], where we follow their approach and set $\mathcal{M} = 5\mathcal{P}$.

The first two terms, $\mathcal{L}_{\text{events}}$ and $\mathcal{L}_{\text{nonevents}}$, were proposed in DyRep [14] and we use them to train our baseline models. The KL divergence term, adopted from NRI [15] to train our LDG models, regularizes the model to align predicted $q_\phi(S|Z)$ and prior $p_\theta(S)$ distributions of attention over edges. Here, $p_\theta(S)$ can, for example, be defined as $[\theta_1, \theta_2, \ldots, \theta_r]$ in case of $r$ edge types.

Following [15], we consider uniform and sparse priors. In the uniform case, $\theta_i = 1/r$, $i = 1, \ldots, r$, such that the KL term becomes the sum of entropies $H$ over the events $p = [1, \ldots, \mathcal{P}]$ and over the generated edges excluding self-loops ($u \neq v$):

$$\text{KL}[q_\phi(S|Z)||p_\theta(S)] = -\sum_p \sum_{u \neq v} H_\phi^p, \tag{14}$$

where entropy is defined as a sum over edge types $r$: $H_\phi^p = -\sum q_\phi^p \log q_\phi^p$ and $q_\phi^p$ denotes distribution $q_\phi(S_{u,v}^{t_p}|Z^{t_p-1})$.

In case of sparse $p_\theta(S)$ and, for instance, $r = 2$ edge types, we set $p_\theta(S) = [0.90, 0.05, 0.05]$, meaning that we generate $r + 1$ edges, but do not use the non-edge type corresponding to high probability, and leave only $r$ sparse edges. In this case, the KL term becomes:

$$\text{KL}[q_\phi(S|Z)||p_\theta(S)] = -\sum_p \sum_{u \neq v} (H_\phi^p - H_{\phi,q}^p), \tag{15}$$

where $H_{\phi,q}^p = -\sum q_\phi^p \log p_\theta(S)$. During training, we update $S_{u,v}^{t_p}$ after each $p$-th event and backpropagate through the entire sequence in a minibatch. To backpropagate through the process of sampling discrete edge values, we use the Gumbel reparametrization [43], as in [15].

**Table 2. Datasets statistics used in experiments.**

| DATASET | SOCIAL EVOLUTION | GITHUB |
|---|---|---|
| # NODES | 83 | 284 |
| # INITIAL ASSOCIATIONS | 575 | 149 |
| # FINAL ASSOCIATIONS | 708 | 710 |
| # TRAIN COMM EVENTS | 43,469 | 10,604 |
| # TRAIN ASSOC EVENTS | 365 | 1,040 |
| # TEST COMM EVENTS | 10,462 | 8,667 |
| # TEST ASSOC EVENTS | 73 | 415 |

## 4 Experiments

### 4.1 Datasets

We evaluate our model on the link (event) prediction task using two datasets (Table 2). Source code is available at https://github.com/uoguelph-mlrg/LDG.

**Social Evolution [16]**. This dataset consists of over one million events $o^t = (u, v, \tau, k)$. We preprocess this dataset in a way similar to [14]. A communication event ($k = 1$) is represented by the sending of an SMS message, or a PROXIMITY or CALL event from node $u$ to node $v$; an association event ($k = 0$) is a CLOSEFRIEND record between the nodes. We also experiment with other associative connections (Fig 5). As PROXIMITY events are noisy, we filter them by the probability that the event occurred, which is available in the dataset annotations. The filtered dataset on which we report results includes 83 nodes with around 43k training and 10k test communication events. As in [14], we use events from September 2008 to April 2009 for training, and from May to June 2009 for testing.

**GitHub**. This dataset is provided by GitHub Archive and compared to Social Evolution is a very large network with sparse association and communication events. To learn our model we expect frequent interactions between nodes, therefore we extract a dense subnetwork, where each user initiated at least 200 communication and 7 association events during the year of 2013. Similarly to [14], we consider FOLLOW events in 2011-2012 as initial associations, but to allow more dense communications we consider more event types in addition to WATCH and STAR: FORK, PUSH, ISSUES, ISSUECOMMENT, PULLREQUEST, COMMIT. This results in a dataset of 284 nodes and around 10k training events (from December to August 2013) and 8k test events (from September to December 2013).

We evaluate models only on communication events, since the number of association events is small, but we use both for training. At test time, given tuple $(u, \tau, k)$, we compute the conditional density of $u$ with all other nodes and rank them [14]. We report Mean Average Ranking (MAR) and HIST@10: the proportion of times that a test tuple appears in the top 10.

### 4.2 Implementation details

We train models with the Adam optimizer [44], with a learning rate of $2\times10^{-4}$, minibatch size $\mathcal{P} = 200$ events, and $d = 32$ hidden units per layer, including those in the encoder (7)–(10). We consider two priors, $p_\theta(S)$, to train the encoder: uniform and sparse with $r = 2$ edge types in each case. For the sparse case, we generate $r + 1$ edges, but do not use the non-edge type corresponding to high probability and leave only $r$ sparse edges. We run each experiment 10 times and report the average and standard deviation of MAR and HIST@10 in Table 3. We train for 5 epochs with early stopping. To run experiments with random graphs, we generate $S$ once in the beginning and keep it fixed during training. For the models with learned temporal

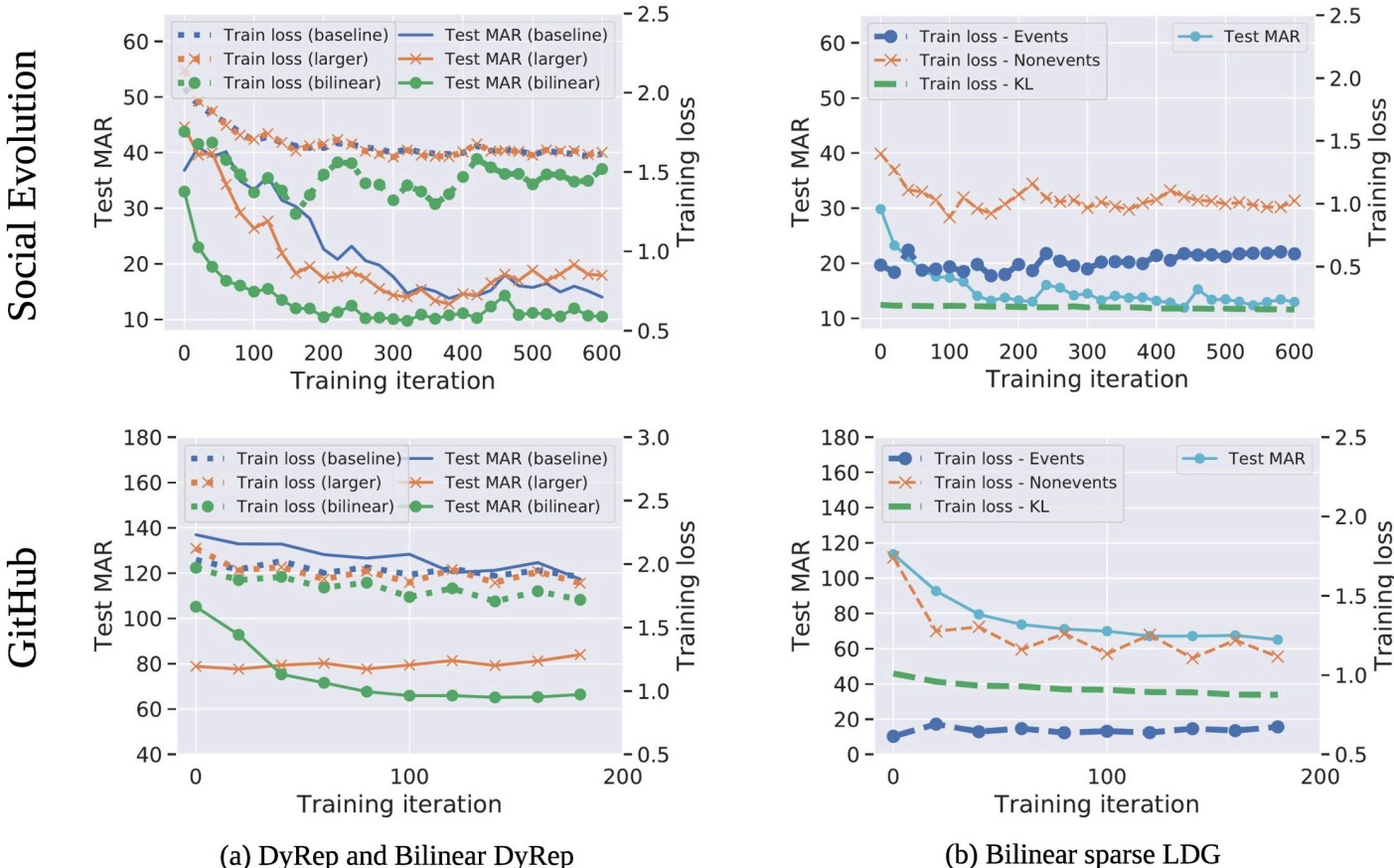

**Fig 5. Predicting links by leveraging training data statistics without any learning ("no learn") turned out to be a strong baseline.** We compare it to learned models with different human-specified graphs used for associations. Here, for the Social Evolution dataset the abbreviations are following: BLOG: BlogLivejournalTwitter, CF: CloseFriend, FB: FacebookAllTaggedPhotos, POL: PoliticalDiscussant, SOC: SocializeTwicePerWeek, RANDOM: Random association graph.

attention, we use random attention values for initialization, which are then updated during training (Fig 6).

We also train two simple baselines based on GCN [8] and GAT [45]. Both of them have three graph convolutional layers with 32 hidden units in each layer and the GAT has 4 attention heads. To train these baselines, in the first training iteration we propagate the initial node embeddings through the GCN or GAT using the association graph at the current timestamp obtaining node embeddings $\mathbf{z}$. We then use $\mathbf{z}$ to compute intensity $\lambda$ for all events in the mini-batch according to (5) or (12) and afterwards the baseline DyRep loss (the first two terms in (13)). For the following training iteration, the input to the GCN or GAT are the output embeddings from the previous training iteration, which allows the embeddings to evolve during training. The main limitation of theses baselines compared to DyRep or LDG is that the communication events are only used to compute $\lambda$ and do not define the node propagation scheme neither during training or testing.

## 4.3 Quantitative evaluation

On Social Evolution, we report results of the baseline DyRep with different human-specified associations and compare them to the models with learned temporal attention (LDG) (Table 3, Fig 5). Models with learned attention perform better than most of the human-specified

**Table 3. Results on the Social Evolution and GitHub datasets in terms of MAR and HITS@10 metrics.** We proposed models with bilinear interactions and learned temporal attention. Bolded results denote best performance for each dataset. Comparison to other baselines and ablation studies are presented in Tables 4 and 5.

| | MODEL | LEARNED ATTENTION | MAR ↓ | | HITS@10 ↑ | |
|---|---|---|---|---|---|---|
| | | | CONCAT | BILINEAR | CONCAT | BILINEAR |
| SOCIAL EVOLUTION | GCN (CLOSEFRIEND) | ✗ | 27.1 ± 0.6 | 29.4 ± 2.6 | 0.42 ± 0.01 | 0.33 ± 0.05 |
| | GAT (CLOSEFRIEND) | ✗ | 48.2 ± 5.0 | 52.4 ± 2.6 | 0.09 ± 0.04 | 0.07 ± 0.02 |
| | DYREP (CLOSEFRIEND) | ✗ | 16.0 ± 3.0 | **11.0 ± 1.2** | 0.47 ± 0.05 | **0.59 ± 0.06** |
| | DYREP (FB) | ✗ | 20.7 ± 5.8 | 15.0 ± 2.0 | 0.29 ± 0.21 | 0.38 ± 0.14 |
| | LDG (RANDOM, UNIFORM) | ✗ | 19.5 ± 4.9 | 16.0 ± 3.3 | 0.28 ± 0.19 | 0.35 ± 0.17 |
| | LDG (RANDOM, SPARSE) | ✗ | 21.2 ± 4.4 | 17.1 ± 2.6 | 0.26 ± 0.10 | 0.37 ± 0.08 |
| | LDG (LEARNED, UNIFORM) | ✓ | 22.6 ± 6.1 | 16.9 ± 1.1 | 0.18 ± 0.09 | 0.37 ± 0.06 |
| | LDG (LEARNED, SPARSE) | ✓ | 17.0 ± 5.8 | 12.7 ± 0.9 | 0.37 ± 0.14 | 0.50 ± 0.06 |
| GITHUB | GCN (FOLLOW) | ✗ | 53.5 ± 1.9 | 53.8 ± 0.3 | 0.36 ± 0.00 | 0.36 ± 0.00 |
| | GAT (FOLLOW) | ✗ | 107.1 ± 3.6 | 111.5 ± 3.9 | 0.21 ± 0.02 | 0.20 ± 0.01 |
| | DYREP (FOLLOW) | ✗ | 100.3 ± 10 | 73.8 ± 5.0 | 0.187 ± 0.01 | 0.221 ± 0.02 |
| | LDG (RANDOM, UNIFORM) | ✗ | 90.3 ± 17.1 | 68.7 ± 3.3 | 0.21 ± 0.02 | 0.24 ± 0.02 |
| | LDG (RANDOM, SPARSE) | ✗ | 95.4 ± 14.9 | 71.6 ± 3.9 | 0.20 ± 0.01 | 0.23 ± 0.01 |
| | LDG (LEARNED, UNIFORM) | ✓ | 92.1 ± 15.1 | 66.6 ± 3.6 | 0.20 ± 0.02 | 0.27 ± 0.03 |
| | LDG (LEARNED, SPARSE) | ✓ | 90.9 ± 16.8 | 67.3 ± 3.5 | 0.21 ± 0.03 | 0.28 ± 0.03 |
| | LDG (LEARNED, SPARSE) + FREQ | ✓ | 47.8 ± 5.7 | **43.0 ± 2.8** | 0.48 ± 0.03 | **0.50 ± 0.02** |

associations, further confirming the finding from [15] that the underlying graph can be suboptimal. However, these models are still slightly worse compared to CLOSEFRIEND, which means that this graph is a strong prior. We also show that bilinear DyRep and LDG models consistently improve results and exhibit better training behaviour, including when compared to a larger linear model with an equivalent number of parameters (Fig 7).

Both DyRep and LDG also outperform simple GCN and GAT baselines, which confirms the importance of modeling the dynamics of both association and communication events. Among these baselines, GAT shows significantly worse results. This can be explained by the difficulty of training attention heads in the GAT. These baselines also often diverge when combined with the bilinear intensity function and so we had to apply additional squashing of

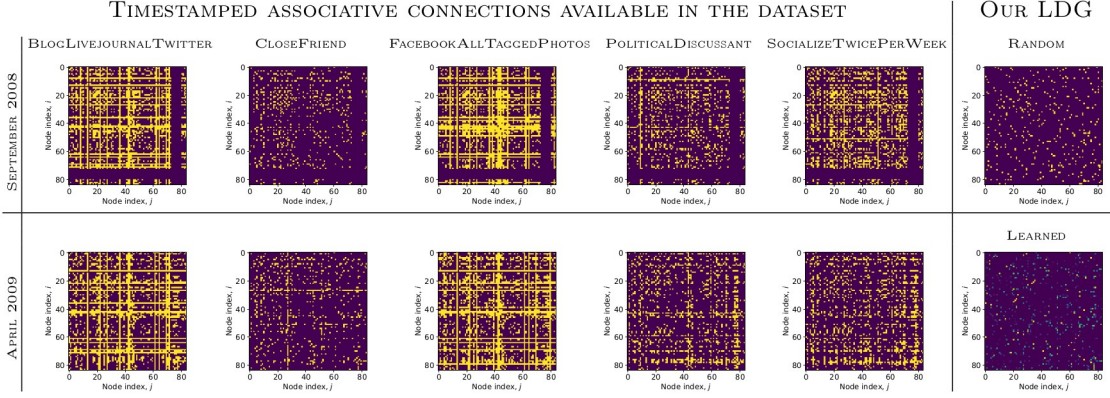

**Fig 6. An adjacency matrix of the latent dynamic graph (LDG), $S^t$, for one of the sparse edge types generated at the end of training, compared to randomly initialized $S^t$ and associative connections available in the Social Evolution dataset at the beginning (top row) and end of training (bottom row).** A quantitative comparison is presented in Table 3.

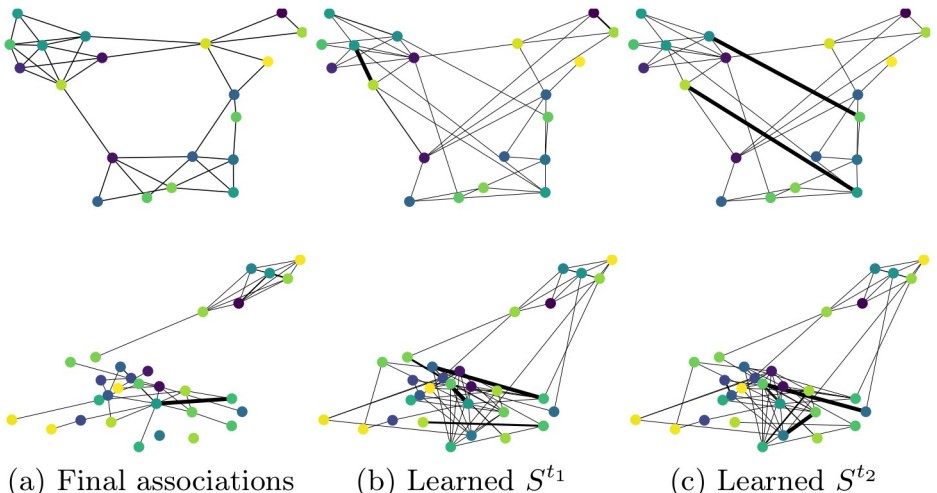

(a) Final associations (b) Learned $S^{t_1}$ (c) Learned $S^{t_2}$

**Fig 7.** (a) Training curves and test MAR for **baseline DyRep, bilinear DyRep**, and **larger baseline DyRep**, with a number of trainable parameters equal to the bilinear DyRep. (b) Training curves and test MAR for the **bilinear LDG with sparse prior**, showing that all three components of the loss (see (13)) generally decrease or plateau, reducing test MAR.

embeddings (see our implementation for full details). Therefore, in these cases, bilinear models underperform.

GitHub is a more challenging dataset with more nodes and complex interactions, so that the baseline DyRep model has a very large MAR of 100 (random predictions have the MAR of around 140). On this dataset, bilinear DyRep and LDG models provide a much larger gain, while models with learned attention further improve results. The baseline DyRep model using FOLLOW associations is performing poorly, even compared to our model with randomly initialized graphs. These results imply that we should either carefully design graphs or learn them jointly with the rest of the model as we do in our work.

Surprisingly, on this dataset, the GCN baseline outperform DyRep and LDG. This can be due to a certain artifact in the dataset or that there exists a simple pattern of events, and more complex models cannot capture them. Note that while GCN is better than LDG in this case, the GCN model leverages the ground truth association graph, which is unavailable to the LDG.

We compare our implementation of DyRep to [14] and other static and dynamic models (Table 4). Our results are worse than those reported in [14] due to the possible implementation differences (no source code of [14] is available) and different preprocessing steps of the Social Evolution dataset. However, we highlight that our results are still significantly better than other baselines.

**Table 4. Comparison of our DyRep implementation to other baselines on the Social Evolution dataset.**

| MODEL | MAR ↓ | HITS@10 ↑ |
|---|---|---|
| GRAPHSAGE [14, 46] | 37.0[†] | 0.12[†] |
| KNOWEVOLVE [14, 47] | 30.0[†] | 0.25[†] |
| DYNGEM [14, 28] | 52.0[†] | 0.01[†] |
| DYREP [14] | 6.0[†] | 0.95[†] |
| DYREP (OUR IMPLEMENTATION) | 16.0 | 0.47 |

[†]According to the results shown on the plots in [14].

**Table 5. The effect of the bilinear transformation when used in different components of our LDG model on the Social Evolution dataset.**

| MODEL | MAR ↓ | HITS@10 ↑ |
|---|---|---|
| DYREP | 16.0 | 0.47 |
| DYREP+BILINEAR λ | 11.0 | 0.59 |
| LDG | 17.0 | 0.37 |
| LDG+BILINEAR NRI ((7))-(10) | 17.9 | 0.38 |
| LDG+BILINEAR Ω (12) | 13.5 | 0.48 |
| LDG+BILINEAR Ω AND NRI | 12.7 | 0.50 |

We also perform additional ablation studies of our LDG (Table 5). In particular, we report the results of using the bilinear transformation only for the encoder (NRI) components or only for computing the conditional intensity. The results suggest a more important role of the latter. While using the bilinear form only for the NRI component degrades the performance, combining it with the bilinear intensity function leads to the best results. The results of our LDG without or with the bilinear transformation on the GitHub dataset are presented in Table 3.

## 4.4 Interpretability of learned attention

While the models with a uniform prior have better test performance than those with a sparse prior in some cases, sparse attention is typically more interpretable. This is because the model is forced to infer only a few edges, which must be strong since that subset defines how node features propagate (Table 6). In addition, relationships between people in the dataset tend to be sparse. To estimate agreement of our learned temporal attention matrix with the underlying association connections, we take the matrix $S^t$ generated after the last event in the training set and compute the area under the ROC curve (AUC) between $S^t$ and each of the associative connections present in the dataset.

These associations evolve over time, so we consider initial and final associations corresponding to the first and last training events. AUC is used as opposed to other metrics, such as accuracy, to take into account sparsity of true positive edges, as accuracy would give overoptimistic estimates. We observe that LDG learns a graph most similar to CLOSEFRIEND. This is an interesting phenomenon, given that we only observe how nodes communicate through many events between non-friends. Thus, the learned temporal attention matrix is capturing information related to the associative connections.

**Table 6. Edge analysis for the LDG model with a learned graph using the area under the ROC curve (AUC, %); random chance AUC = 50%.** CLOSEFRIEND is highlighted as the relationship closest to our learned graph.

| ASSOCIATIVE CONNECTION | INITIAL | | FINAL | |
|---|---|---|---|---|
| | UNIFORM | SPARSE | UNIFORM | SPARSE |
| BLOGLIVEJOURNALTWITTER | 53 | 57 | 57 | 69 |
| CLOSEFRIEND | 65 | 69 | 76 | 84 |
| FACEBOOKALLTAGGEDPHOTOS | 55 | 57 | 58 | 62 |
| POLITICALDISCUSSANT | 59 | 63 | 61 | 68 |
| SOCIALIZETWICEPERWEEK | 60 | 66 | 63 | 70 |
| GITHUB FOLLOW | 79 | 80 | 85 | 86 |

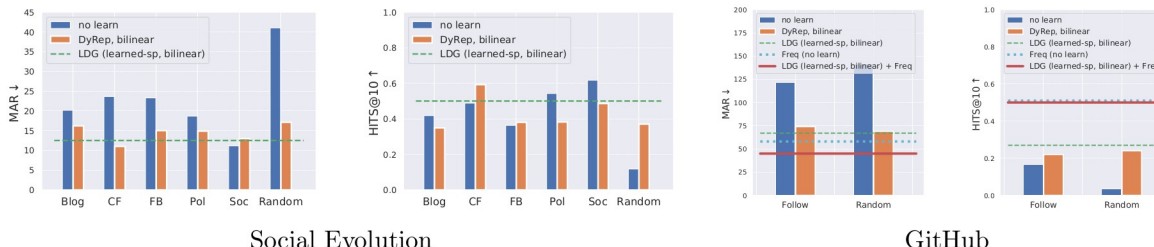

Social Evolution GitHub

**Fig 8. Final associations of the subgraphs for the Social Evolution (top row) and GitHub (bottom row) datasets compared to attention values at different time steps selected randomly.**

We also visualize temporal attention sliced at different time steps and can observe that the model generally learns structure similar to human-specified associations (Fig 8). However, attention can evolve much faster than associations, which makes it hard to analyze in detail. Node embeddings of bilinear models tend to form more distinct clusters, with frequently communicating nodes generally residing closer to each other after training (Fig 9). We notice bilinear models tend to group nodes in more distinct clusters. Also, using the random edges

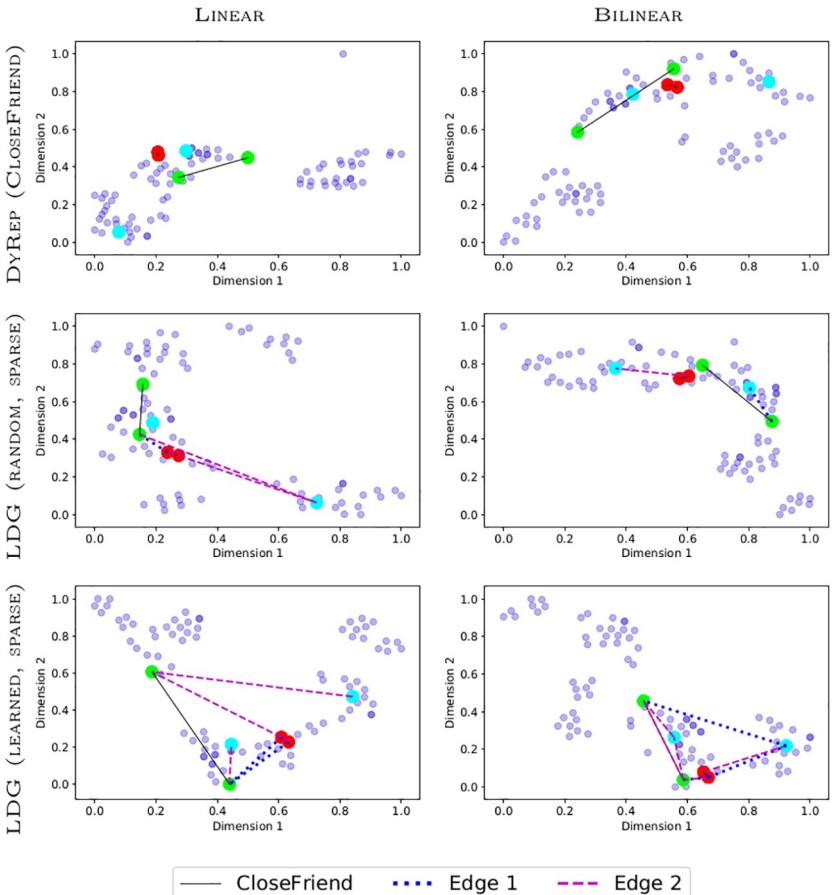

**Fig 9. tSNE node embeddings after training (coordinates are scaled for visualization) on the Social Evolution dataset.** Lines denote associative or sampled edges. Darker points denote overlapping nodes. Red, green, and cyan nodes correspond to the three most frequently communicating pairs of nodes, respectively.

approach clusters nodes well and embeds frequently communicating nodes close together, because the embeddings are mainly affected by the dynamics of communication events.

## 4.5 Leveraging the dataset bias

While there can be complex interactions between nodes, some nodes tend to communicate only with certain nodes, which can create a strong bias. To understand this, we report results on the test set, which were obtained simply by computing statistics from the training set (Fig 5).

For example, to predict a link for node $u$ at time $\tau$, we randomly sample node $v$ from those associated with $u$. In the RANDOM case, we sample node $v$ from all nodes, except for $u$. This way we achieve high performance in some cases, e.g., MAR = 11 by exploiting SOCIALIZETWICEPER-WEEK, which is equivalent to learned DyRep. This result aligns with our intuition that people who socialize with each other tend to communicate more, whereas such associations as friends tend to be stronger, longer term relationships that do not necessary involve frequent communications.

Another bias present in the datasets is the frequency bias, existing in many domains, e.g. visual relationship detection [48]. In this case, to predict a link for node $u$ at time $\tau$ we can predict the node with which it had most of communications in the training set. On GitHub this creates a very strong bias with performance of MAR = 58. We combine this bias with our LDG model by averaging model predictions with the frequency distribution and achieve our best performance of MAR = 45. Note that this bias should be used carefully as it may not generalize to another test set. A good example is the Social Evolution dataset, where communications between just 3-4 nodes correspond to more than 90% of all events, so that the rank of 1-2 can be easily achieved. However, such a model would ignore potentially changing dynamics of events at test time. Therefore, in this case the frequency bias is exploiting the limitation of the dataset and the result would be overoptimistic. In contrast the DyRep model and our LDG model allow node embeddings to evolve over time based on current dynamics of events, so that it can adapt to such changes.

## 5 Conclusion

We introduced a novel model extending DyRep and NRI for dynamic link prediction. We found that, in many cases, attention learned using our model can be advantageous compared to attention computed based on the human-specified graph, which can be suboptimal and expensive to produce. Furthermore, we showed that bilinear layers can capture essential pairwise interactions outperforming a more common concatenation-based layer in all evaluated cases.

## Acknowledgments

The authors thank Elahe Ghalebi and Brittany Reiche for their helpful comments.

## Author Contributions

**Conceptualization:** Graham W. Taylor.

**Data curation:** Boris Knyazev.

**Methodology:** Carolyn Augusta.

**Software:** Boris Knyazev.

**Supervision:** Graham W. Taylor.

**Writing – original draft:** Boris Knyazev, Carolyn Augusta.

**Writing – review & editing:** Boris Knyazev, Carolyn Augusta, Graham W. Taylor.

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
