## [Decision Letter · Decision Letter 0]

5 Oct 2020

PONE-D-20-21179

Learning temporal attention in dynamic graphs with bilinear interactions

PLOS ONE

Dear Dr. Augusta,

Thank you for submitting your manuscript to PLOS ONE. After careful consideration, we feel that it has merit but does not fully meet PLOS ONE’s publication criteria as it currently stands. Therefore, we invite you to submit a revised version of the manuscript that addresses the points raised during the review process.

We look forward to receiving your revised manuscript.

Kind regards,

Chi Ho Yeung

Academic Editor

PLOS ONE

Journal Requirements:

2BK is funded by the Mila internship, the Vector Institute, the University of Guelph and DARPA (FA8750-17-C-0115). CA is funded by the University of Saskatchewan. GWT is funded by CIFAR, Canada Research Chairs and the University of Guelph. For a portion of time during this study, GWT received salary from Google. The views, opinions and/or findings expressed are those of the authors and should not be interpreted as representing the official views or policies of the Department of Defense or the U.S. Government. The authors also acknowledge equipment support from Canada Foundation for Innovation. Resources used in preparing this research were provided, in part, by the Province of Ontario, the Government of Canada through CIFAR, and companies sponsoring the Vector Institute: " ext-link-type="uri" xlink:type="simple">http://www.vectorinstitute.ai/#partners."

We note that one or more of the authors are/were employed by a commercial company: Google.

2.1. Please provide an amended Funding Statement declaring this commercial affiliation, as well as a statement regarding the Role of Funders in your study. If the funding organization did not play a role in the study design, data collection and analysis, decision to publish, or preparation of the manuscript and only provided financial support in the form of authors' salaries and/or research materials, please review your statements relating to the author contributions, and ensure you have specifically and accurately indicated the role(s) that these authors had in your study. You can update author roles in the Author Contributions section of the online submission form.

2.2. Please also provide an updated Competing Interests Statement declaring this commercial affiliation along with any other relevant declarations relating to employment, consultancy, patents, products in development, or marketed products, etc.  

3. We noted in your submission details that a portion of your manuscript may have been presented or published elsewhere.

" The data have been published elsewhere and are publicly available. We do not create data, but rather use these two publicly available data sets.

Madan A, Cebrian M, Moturu S, Farrahi K, et al. Sensing the \\health state" of a

community. IEEE Pervasive Computing. 2012;11(4):36{45."

Reviewers' comments:

Reviewer's Responses to Questions

**Comments to the Author**

1. Is the manuscript technically sound, and do the data support the conclusions?

Reviewer #1: Yes

Reviewer #2: Yes

2. Has the statistical analysis been performed appropriately and rigorously? 

Reviewer #1: Yes

Reviewer #2: Yes

3. Have the authors made all data underlying the findings in their manuscript fully available?

Reviewer #1: Yes

Reviewer #2: Yes

4. Is the manuscript presented in an intelligible fashion and written in standard English?

Reviewer #1: Yes

Reviewer #2: Yes

5. Review Comments to the Author

Reviewer #1: The paper is interesting and well written. They have good results and they are presented well.

Just a few minor points:

1) The task you are focusing on is event prediction, but you don't really state it clearly (casually mentioned in the DyRep backgroup part). Please state it more clearly for clearity.

2) There is an issue with the figures, they are in the end of the paper instead of embedded in it as I assume the authors intended.

3) minor typo in dimension, line 205

Reviewer #2: The manuscript focuses on reasoning dynamic graph data and proposed a model called Latent Dynamic Graph (LDG). The authors first introduced Neural Relational Inference (NRI) into the existing DyREP [1] model for link prediction on dynamic graphs. Then they replace the concatenation based layers in DyREP with bilinear layers. The experiment result shows that in many cases, during model training, using NRI, i.e. discarding human-specified graph and letting the model capture the graph structure implicitly, can be better than using human-specified graph. Moreover, bilinear layers are superior to simple concatenation in capturing pairwise interaction.

Strength:

1. The combination of NRI and DyREP makes the learning process independent of human-specified graph, and the proposed model LDG can learn a similar attention structure to the ground truth association graph, as stated in Fig. 8. It is novel to apply this method to improve the DyREP model.

2. By introducing the bilinear encoder and the bilinear intensity function, the performance of LDG is better than baseline method DyREP in almost all cases stated in Table 3.

Weakness:

1. The number of baseline models is too small. It is better to include more results of other models doing dynamic link prediction on the same dataset. In DyREP’s paper, Know-Evolve [2], DynGEM [3], GraphSage [4] and GAT [5] are used as baselines. It might strengthen this submission’s contributions if the authors could compare one or two more baselines. It might strengthen this submission’s contributions if the authors could compare one or two more baselines. Besides, a recent model architecture for link forecasting on dynamic and multi-relational graphs is proposed in [6], which is missing in the related work.

2. To show the superiority of bilinear layers, it is better to do an ablation study regarding it. For example, the authors can first give the experiment result of LDG by maintaining concatenation in equation (8), (10), (12). Then replace these concatenation with bilinear layers in equation (8), (10), (12), and report the experiment result as well.

3. Similar to point 2, the authors can also do another ablation study to show how NRI affects the performance. For example, the authors can first give the experiment result of LDG by maintaining concatenation in equation (8), (10), (12), and then compare this result with the performance of DyREP. To summarize, I recommend this paper to be accepted. However, there is still a lot of work to do to improve it. The most important point is to compare LDG with more models

from different angles.

4. Current datasets contain only thousands of nodes, which might be impractical for real applications. The reviewers understand that the authors mainly aim to compare and beat the DyRep model, therefore, it would be great if the authors could implement and compare their methods on larger datasets, but not necessary.

[1] DYREP: LEARNING REPRESENTATIONS OVER DYNAMIC GRAPHS

[2] Know-evolve: Deep temporal reasoning for dynamic knowledge graphs

[3] Dyngem: Deep embedding method for dynamic graphs

[4] Inductive representation learning on large graphs

[5] Graph attention networks.

[6] Graph Hawkes Neural Network for Forecasting on Temporal Knowledge Graphs

6. PLOS authors have the option to publish the peer review history of their article (what does this mean?). If published, this will include your full peer review and any attached files.

Reviewer #1: No

Reviewer #2: No

---

## [Author Response · Author response to Decision Letter 0]

25 Jan 2021

We also attach response.pdf containing the same information as below.

We thank the reviewers for the useful feedback and comments.

Below is the list of our changes.

Reviewer #1: 

1) The task you are focusing on is event prediction, but you don't really state it clearly (casually mentioned in the DyRep backgroup part). Please state it more clearly for clearity.

We added the statement in the beginning of the Experiments section.

2) There is an issue with the figures, they are in the end of the paper instead of embedded in it as I assume the authors intended.

We followed the submission instructions and did not embed the figures in the manuscript.

3) minor typo in dimension, line 205

Fixed.

Reviewer #2: 

1. The number of baseline models is too small. It is better to include more results of other models doing dynamic link prediction on the same dataset. In DyREP’s paper, Know-Evolve [2], DynGEM [3], GraphSage [4] and GAT [5] are used as baselines. It might strengthen this submission’s contributions if the authors could compare one or two more baselines. It might strengthen this submission’s contributions if the authors could compare one or two more baselines. Besides, a recent model architecture for link forecasting on dynamic and multi-relational graphs is proposed in [6], which is missing in the related work.

We added the results of GAT and GCN baselines implemented and evaluated using our codebase to Table 3. We also added Table 4 comparing DyRep to Know-Evolve, DynGEM and GraphSage using the results reported in the DyRep paper. We also added a description of our GAT/GCN baselines in Implementation Details and additional discussion of results in Quantitative Evaluation.

We added a paragraph to Related Work discussing [6].

2. To show the superiority of bilinear layers, it is better to do an ablation study regarding it. For example, the authors can first give the experiment result of LDG by maintaining concatenation in equation (8), (10), (12). Then replace these concatenation with bilinear layers in equation (8), (10), (12), and report the experiment result as well.

We added Table 5 with these results.

3. Similar to point 2, the authors can also do another ablation study to show how NRI affects the performance. For example, the authors can first give the experiment result of LDG by maintaining concatenation in equation (8), (10), (12), and then compare this result with the performance of DyREP. To summarize, I recommend this paper to be accepted. However, there is still a lot of work to do to improve it. The most important point is to compare LDG with more models from different angles.

This seems to be very similar to the point above. We hope the added Table 5 together with existing Table 3 provide an evaluation of LDG from different angles.

4. Current datasets contain only thousands of nodes, which might be impractical for real applications. The reviewers understand that the authors mainly aim to compare and beat the DyRep model, therefore, it would be great if the authors could implement and compare their methods on larger datasets, but not necessary.

We agree with this point, but this requires substantial engineering efforts, which we could not allocate in the time limit provided. In general, using our LDG model on larger datasets is possible with more GPU memory and/or more efficient implementation. In this work, we train our models on a single NVIDIA 1080/2080 Ti with 11GB of GPU memory.

[1] DYREP: LEARNING REPRESENTATIONS OVER DYNAMIC GRAPHS

[2] Know-evolve: Deep temporal reasoning for dynamic knowledge graphs

[3] Dyngem: Deep embedding method for dynamic graphs

[4] Inductive representation learning on large graphs

[5] Graph attention networks.

[6] Graph Hawkes Neural Network for Forecasting on Temporal Knowledge Graphs

---

## [Editor Report · Decision Letter 1]

17 Feb 2021

Learning temporal attention in dynamic graphs with bilinear interactions

PONE-D-20-21179R1

Dear Dr. Augusta,

We’re pleased to inform you that your manuscript has been judged scientifically suitable for publication and will be formally accepted for publication once it meets all outstanding technical requirements.

Kind regards,

Chi Ho Yeung

Academic Editor

PLOS ONE

Additional Editor Comments (optional):

The authors have well addressed the questions raised by both reviewers, I therefore recommend the manuscript to be accepted for publication.
---

## [Editor Report · Acceptance letter]

22 Feb 2021

PONE-D-20-21179R1 

Learning temporal attention in dynamic graphs with bilinear interactions 

Dear Dr. Augusta:

I'm pleased to inform you that your manuscript has been deemed suitable for publication in PLOS ONE. Congratulations! Your manuscript is now with our production department. 

Kind regards, 

on behalf of

Dr. Chi Ho Yeung 

Academic Editor

PLOS ONE